# The Perfect Storm: Applying the Multiple Streams Framework to Understand the Adoption of a WHO Framework Convention on Tobacco Control-Based Policy in Mexico

**DOI:** 10.3390/ijerph21070917

**Published:** 2024-07-13

**Authors:** Eric Crosbie, Sara Perez, Adriana Rocha Camarena, Valentina Ochoa Vivanco, Gianella Severini, Patricia Gutkowski, Patricia Sosa, Ernesto M. Sebrié

**Affiliations:** 1School of Public Health, University of Nevada Reno, Reno, NV 89557, USA; saraperez@unr.edu; 2Ozmen Institute for Global Studies, University of Nevada Reno, Reno, NV 89557, USA; 3México SaludHable, Mexico City 01000, Mexico; adriana.rocha.camarena@gmail.com; 4Campaign for Tobacco-Free Kids, Washington, DC 20005, USA; vochoa@advocacyincubator.org (V.O.V.); gseverini@tobaccofreekids.org (G.S.); pgutkowski@tobaccofreekids.org (P.G.); psosa@advocacyincubator.org (P.S.); esebrie@tobaccofreekids.org (E.M.S.)

**Keywords:** health advocacy, tobacco control, tobacco industry, health policy, Mexico

## Abstract

Background: The aim of this study was to document how Mexico adopted a WHO Framework Convention on Tobacco Control (FCTC)-based national tobacco control law. Methods: We analyzed publicly available documents and interviewed 14 key stakeholders. We applied the Multiple Streams Framework (MSF) to analyze these findings. Results: Previous attempts to approve comprehensive FCTC-based initiatives failed due to a lack of political will, the tobacco industry’s close connections to policymakers, and a lack of health advocacy coordination. Applying the MSF reveals increased attention towards collecting and sharing data to frame the severity of the problem (problem stream). The expansion of a coordinated health advocacy coalition and activities led to increased support for desired FCTC policy solutions (policy stream). The election of President López Obrador and legislative changes led to a deep renewed focus on tobacco control (politics stream). These three streams converged to create a policy window to secure a strong FCTC-based initiative on the political agenda that was ultimately passed. Conclusions: The Mexican experience illustrates the importance of continued health advocacy and political will in adopting FCTC-based policies. Other countries should follow Mexico’s lead by collecting and sharing data through coordinating efforts in order to be prepared to seize political opportunity windows when strong political will is present.

## 1. Introduction

Tobacco use is the most preventable cause of disability and death worldwide as more than eight million people die annually, and 80% of these deaths occur in low- and middle-income countries (LMICs) [1]. In 2003, World Health Organization (WHO) Member States adopted the WHO Framework Convention on Tobacco Control (FCTC), a global health treaty that places obligations on parties to reduce the supply and demand of tobacco [2]. Since coming into effect in 2005, the WHO FCTC has helped accelerate the adoption of evidence-based best practices aimed at reducing tobacco use, nicotine addiction, and exposure to secondhand tobacco smoke including establishing smoke-free public places and workplaces (Article 8), banning tobacco advertising, promotion, and sponsorship (TAPS) (Article 13), and requiring pictorial health warning labels (HWLs) (Article 11) [3,4,5].

The adoption of WHO FCTC-based policies has significantly progressed in the Latin America and Caribbean (LAC) region. As of May 2024, all 33 LAC countries, except 4 (Argentina, Cuba, Dominican Republic, and Haiti), have ratified the FCTC [1]. Twenty-two countries require pictorial HWLs printed on cigarette packages [6], nine have adopted comprehensive TAPS bans [7], and twenty-six countries, including the entire sub-region of South America, has adopted 100% smoke-free indoor public places [8].

Despite this success, the focus of the literature analyzing the adoption of WHO FCTC policies has primarily focused on high-income countries (HICs) in analyzing facilitators and barriers to policy adoption [9]. While recently published studies have begun to assess policy adoption and implementation of WHO FCTC-based best practices in LMICs, significant knowledge gaps remain [10,11]. In particular, there is a lack of analyzing LMIC case studies to understand how tobacco control initiatives reach policy agendas and are ultimately passed into law. This is especially the case in Mexico, which is the second most populous LAC country, home to over 130 million people. In 2008, the Mexican government adopted a weak federal tobacco control law, the General Law for Tobacco Control (LGCT, la Ley General para el Control del Tabaco) that, among other measures, required designated smoking areas in public places [12]. Following a decade of numerous failed attempts to either pass a new FCTC-based law or amend the LGCT, on 17 February 2022, an amendment to the LGCT was officially published, which established 100% smoke-free public places including outdoor spaces and for electronic cigarettes and heated tobacco products, and completely prohibited TAPS [13] (Table 1). Given the repeated failures to amend the LGCT, this represents the first known study that attempts to understand the process of adoption of a WHO FCTC-based national tobacco control law in Mexico. In doing so, this study hopes to provide lessons for other LMICs, especially in LAC, working to adopt policies aligned with the WHO FCTC and its implementation guidelines.

## 2. Materials and Methods

### 2.1. Data Collection

#### Interviews with Key Informants

Between December 2023 and March 2024, we invited via email 24 key stakeholders that had been involved in tobacco control in Mexico between 2008 and 2020, to participate in an in-depth interview. Interviewees were identified through media searches, the authors’ networks, and snowball sampling. Fourteen agreed to be interviewed, six denied our requests, and four never responded after five requests. The interviewees included public health advocates (n = 5), academic researchers (n = 2), Mexican policymakers and government officials (n = 5), and inter-governmental organization officials (n = 2). The interviewees were emailed semi-structured interview questions and agreed through verbal consent to participate in the study in accordance with a protocol approved by the University of Nevada, Reno Committee on Human Research (Appendix A). Interviews were conducted and recorded via Zoom and lasted approximately 45–60 min. Interviewees also provided 14 public documents (e.g., draft legislation, health advocacy reports, press conferences) to provide further documentation and context for this study.

## 3. Data Analysis

Given that Mexico had failed to amend the LGCT for more than a decade, we adopted Kingdon’s Multiple Streams Framework (MSF) to understand how the policy eventually made its way onto the policy agenda setting stage and was ultimately passed [14]. MSF is used to highlight why particular policy issues make the agenda of policymakers while others fail. The MSF hypothesizes that a policy problem is more likely to gain attention from policymakers when it is “coupled” with a policy solution under a supportive political environment [14]. In particular, the MSF examines three semi-independent “streams” that travel through the policy process: the problem stream, policy stream, and politics stream [14]. The problem stream relates to how policymakers, the media, and the public define or frame a problem as this can constrain the types of policy solutions that policymakers consider. This can occur when accumulating factors lead to increased attention to a policy problem, when there is a focusing event (e.g., disaster), or if there appears to be a feasible or effective solution in a given political context. The policy stream focuses on the generation of policy solutions that have risen or are expected to rise on the political agenda. Policy entrepreneurs, or individuals willing “to invest their resources-time, energy, reputation, and sometimes money”, advocate for specific policy solutions and often work to build coalitions that increase support for their desired policy solutions [14]. Finally, the politics stream focuses on the broader socio-political environment where problems and solutions are defined and connected. This can include factors such as public mood, advocacy campaigns, and election results, where policy entrepreneurs can judiciously navigate the political landscape to connect their desired policies with significant problems [14].

Given these three streams, the MSF hypothesis suggests that an “issue’s chances of gaining agenda status dramatically increase when all three streams—problems, policies, and politics—are coupled in a single package” [15]. In other words, the opening of a policy window determines the likelihood that an issue will become a policy agenda item. This mostly occurs when a policy window opens in the problem or politics stream that emphasizes the need for policy action [14]. While policy windows are particularly opportune moments for agenda setting, they are not always pathways for policy change. Instead, a “problem must be coupled with a solution in a way that is attractive and coherent to receptive policymakers and, potentially, the public while the window is open” [16]. Then, policymakers must enter into the policymaking process to formulate, negotiate, and adopt a specific policy [15]. This study analyzes the data through the MSF to understand how initiatives to reform LGCT made their way onto the political agenda and ultimately passed. 

## 4. Results

### 4.1. Previous Efforts Failing to Reform the LGCT

Between 2008 and 2020, over 100 initiatives were introduced to either amend the LGCT or introduce a new tobacco control law. All participants that were interviewed for this study claimed that these failures stemmed from (1) a lack of political will, (2) tobacco industry political influence with policymakers, and (3) a lack of health advocacy coordination [17,18,19,20,21,22,23,24,25,26,27,28,29,30] (Table 2).

### 4.2. Lack of Political Will

Since the LGCT was passed in 2008, the Mexican government was led by President Felipe Calderón (2006–2012) of the conservative right-wing National Action Party (PAN, Partido Acción Nacional) and Enrique Peña Nieto (2012–2018) of the center-right Institutional Revolutionary Party (PRI, Partido Revolucionario Institucional). Between 2008 and 2018, these political parties also made up the majority in both chambers of the National Congress. Several interviewees claimed that both the PAN and PRI favored economic interests over health [17,21,25,27], and some mentioned that even representatives from the Health Minister at times did not want to move any tobacco control initiatives forward [19,24,25]. One health advocate stated that anything that had to do with health hardly came out of the health committee [26]. When there was support from a PAN member of the health committee to amend the LGCT, the Health Minister from the PRI did not support it [23]. Furthermore, there was no political champion to lead tobacco control efforts during this time period [22].

### 4.3. Tobacco Industry Political Influence

Although several policymakers lacked political will to amend the LGCT, the tobacco industry in Mexico played a significant role in influencing these policymakers [17,18,19,23,24,25,26,27,28]. Tobacco companies, including Philip Morris International and British American Tobacco, have been powerful in Mexico [27,28], establishing close relations with policymakers, mostly PAN and PRI policymakers in high profile positions [18]. A couple of health advocates claimed that any progress would be stalled once it reached high levels such as the Health Minister [17], who at times has been caught negotiating and receiving funding and gifts from the industry [24]. Others advocate that the industry was opportune to consistently influence committee presidents to block any tobacco control initiatives from being discussed so they would not reach the plenary for further discussion [18,23]. When discussion occurred, the industry strategically visited legislators and their legislative advisors to review technical issues to influence them to voice or vote against any proposals so they would not progress out of committee [18,25].

### 4.4. Lack of Health Advocacy Coordination

While local tobacco control groups worked together with international health organizations, several interviewees mentioned that efforts were not as consolidated and coordinated during the late 2000s and 2010s [17,18,24,25,27]. Some health advocates mentioned that previous efforts were ‘isolated’ and did not go anywhere [17]. Other advocates claimed they did not have the physical infrastructure and human resources to carry out particular operations [24]. Furthermore, others claimed that the coordination was more informal, leading to a lack of consensus and urgency in advocating for certain legislative bills [18,25]. 

## 5. Problem Stream

This section of the policymaking process examines how defining and framing the problem evolved over time led to increased attention to a policy problem that policymakers would consider policy solutions (Table 2).

### 5.1. Indicators/Statistics

In defining the severity of the problem of tobacco use in Mexico, health advocates and government officials relied on local and international indicators and statistics. Several interviewees claimed that a lot of support came from annual nationwide surveys on smoking prevalence conducted by the National Institute of Public Health (INSP, Instituto Nacional de Salud Pública). Additionally, the INSP participated in a study with the University of Washington International Health Metric Evaluation program in assessing tobacco control policies in 32 Mexican states and its implications on tobacco consumption and health effects, which helped inform policy debates [31]. This was complemented by the Global Adult Tobacco Survey (GATS), which was conducted three times in Mexico (2009, 2015, and 2023) [17,23]. This provided national representative data to measure the magnitude of smoking, which helped illustrate the lack of progress in areas such as youth smoking [17,18,24,25]. Additionally, the GATS surveys provided significant policy feedback using the WHO MPOWER, a package of six evidence-based demand reduction measures contained in the WHO FCTC, which allowed the advocates to fully monitor the population [22]. These surveys helped define the problem with “great precision” to identify policies that needed to be adopted [22]. Over time, these surveys also highlighted tobacco use consumption state by state [17], and how public support was growing for tobacco control policies such as smoke-free environments and TAPS restrictions [23].

### 5.2. Feedback from Other Policies

In addition to collecting important data and evidence in Mexico, important feedback was given about progress and success occurring in Latin America. Several interviewees claimed that, over time, countries such as Brazil, Uruguay, and Panama adopted WHO FCTC-based policies and experienced great success, decreasing smoking prevalence levels, altering social norms regarding smoking, lowering youth smoking initiation rates, and reducing the burden of death due to tobacco use [17,18,19,24]. Health advocates and government officials in Mexico also attended regional tobacco control conferences in Mexico and abroad to learn about best practices [18,24,25]. These conferences were important networking opportunities that enabled leaders from other countries to visit Mexico to further educate policymakers in Mexico about the progress and success in their own country [24,25]. Over time, this helped build capacity in Mexico [22] but also created increased urgency as Mexico began to lag behind other countries in adopting WHO FCTC-based policies [21,23,24]. For example, by 2021 [8], all of South America had established 100% smoke-free environments, whereas Mexico was still lagging behind [21,23].

### 5.3. Framing the Issue

Given the success in collecting and generating evidence along with important regional feedback, health advocates began to frame the issue more with urgency, targeting industry, youth protection, and adoption of WHO FCTC-based policies. A few health advocates mentioned that for years without data and these successes from other countries, framing tobacco control issues focused more on the smokers and the habit of smoking [17,20,26]. As a result, the focus tended to concentrate heavily on treatment and assisting smokers to quit smoking [17,20,26], yet these efforts were still underfunded and limited at the time. Over time, the framing changed to concentrate more on prevention and risk factors associated with non-communicable diseases. In particular, advocates would connect risk factors with policy interventions and use collected data to illustrate how many lives could be saved if the government adopted smoke-free policies and TAPS bans [23]. These approaches also zeroed in on protecting youth who were more susceptible to industry targeted marketing [21,23,24]. This framing connected with a renewed focus on industry, highlighting industry political and marketing practices [17,19,21]. Given past struggles with the Ministry of Economy and lessons learned elsewhere, advocates also increasingly framed these issues as economic issues (life and medical care costs) [17,23], development issues (meeting United Nations Sustainable Development Goals) [22], and legal issues (Mexico’s constitution Article 4 establishes the right to the protection of health) [17]. This evidence, lessons from other countries, and framing helped generate momentum to amend the LGCT.

## 6. Policy Stream

This section of the policymaking process analyzes the various policy solutions that have risen to the political agenda and how policy entrepreneurs have worked to build coalitions to increase support for amending the LGCT.

### 6.1. Policy Entrepreneurs

Throughout the late 2000s and 2010s, tobacco control coalitions and coordination continued to grow and expand both within the country and regionally. According to some health advocates, the earlier generation of leaders were overcommitted and at times were forced to work in isolation as more emphasis was placed on individual experts to help [17,22,23]. However, this capacity changed when there was more investment in training and projects that came from inter-governmental organizations such as the Pan American Health Organization (PAHO) and international public health organizations such as the Campaign for Tobacco-Free Kids (CTFK) and the International Union Against Tuberculosis and Cancer (the Union) [17,22,23]. This support helped place more emphasis on the group as a transnational tobacco control network and fueled stronger coalition building and coordination among local and national public health groups in Mexico and regionally, producing high levels of cooperation and collaboration [17,22]. This collaboration and coordination continued with the media as, according to one advocate, it “extended to journalists who trusted us and looked for us” [19]. Another health advocate discussed becoming more aggressive in approach, stating that previously, they would not call out political parties, worrying about the consequences, but then experienced greater freedom in pressuring policymakers [23]. Finally, one health advocate stated that even during down times politically, when there was a lack of political will, it was “a period rich in research and collaboration” [24].

Meanwhile, despite failed attempts to amend the LGCT at the national level, important advocacy and policy adoption occurred at the state and local levels throughout the 2010s. Following the Supreme Court’s decision regarding Mexico City’s 2008 smoke-free law to allow subnational governments the authority to adopt stronger policies than the LGCT, some of the largest and most populated states such as Mexico, Tabasco, Tamaulipas, and Oaxaca adopted 100% smoke-free laws [17,22,24]. By 2020, 15 states had adopted 100% smoke-free laws, covering more than 60% of the entire population of Mexico [32]. Several interviewees claimed that this progress at the subnational level helped continue to expand tobacco control coalitions and build momentum to amend the LGCT by using successful local evidence to lobby national policymakers [22]. Others claimed that this positioned health advocates to be more active locally, helping build supportive public health narratives to expand the capacity of journalists in the local media to gain more earned media coverage [17,23]. Furthermore, local health groups such as Salud Justa, CODICE, and Refleacciona helped map out local capabilities and achievements, which in turn helped generate more consensus and expand coalitions across the country, recruiting support from local universities, media companies, and political champions [22,27].

### 6.2. Approaching Policymakers

While there continued to be a lack of political will during the 2010s, tobacco control advocates relentlessly continued to attempt to convince policymakers to advance tobacco control. This was a time period of strategically identifying and cultivating relationships with key policymakers who may support or oppose tobacco control initiatives in the future [24]. As one health advocate explained, “this was a time of sowing seeds and cultivating relationships with key actors in the legislature”, which is “politically profitable as media will seek them out” [18]. Another interviewee mentioned the importance of political mapping and identifying loud opposing voices such as a legislator from Nayarit, a tobacco-growing state in Mexico, that strongly supported tobacco farming and opposed tobacco control initiatives [22]. These efforts were also attempts to gain positioning around the problem and work within the priorities of different political agendas [22].

### 6.3. Policy Alternatives

Throughout the 2010s, health advocates also gained experience in learning which policy solutions were more feasible than others. Given the local success surrounding 100% smoke-free environments and to a lesser extent with TAPS restrictions, several interviewees claimed that this momentum over time had an effect of policymakers being more willing to accept these policy solutions [18,19,24,25,27,28]. One government official noted that “smoke-free became more acceptable due to social norm change as you would now go to a restaurant and increasingly see smokers know to go outside and smoke” [18]. Other interviewees claimed that surveys from the INSP and GATS also showcased how other countries were passing 100% smoke-free laws and TAPS restrictions and experiencing great success [23,24]. Other policy solutions such as tobacco standardized packaging and tobacco flavor bans appeared to meet more resistance [18,19,24,25,27,28]. In particular, some claimed that the industry aggressively argued against these proposals and brought a lot of economic and legal pressure, forcing policymakers and government officials to avoid further discussions for these proposals [18,19,24,25,27,28]. Despite these setbacks, health advocates mentioned that by testing their arguments over many years, they had a better idea of which ones were more persuasive and which helped in terms of positioning tobacco control [23].

## 7. Politics Stream

This section of the policymaking process examines how an election and administrative turnover combined with public health pressure campaigns helped lead to an opening of the political opportunity window to finally reform the LGCT despite continued industry opposition.

### 7.1. Administrative or Legislative Turnover

All interviewees claimed that the election of President Andrés Manuel López Obrador (2018–2024) of the leftist political party the National Regeneration Movement (MORENA, Movimiento Regeneración Nacional) and a majority of MORENA elected officials in 2018 played a critical role in the amendment of the LGCT [17,18,19,20,21,22,23,24,25,26,27,28,29,30]. This elective and administrative change was a key turning point that led to a fundamental shift in prioritizing public health [24,28]. Several interviewees mentioned that political will change was noticeable immediately as there was a greater sense of responsibility [20,24]. According to interviewees, President López Obrador’s leadership and commitment to advancing public health trickled down to all institutions involved in tobacco-related issues, and his political agenda was carried out by policymakers in congress [17,21]. In particular, this change included the leadership of Dr. Hugo López-Gatell, Undersecretariat of Prevention and Health Promotion in the Health Ministry, and later, political champions Dr. Carmen Medel Palma, president of the Chamber of Deputies health committee, and Ernesto Pérez Astorga, member of the Senate Health Committee [19,22]. One government official claimed that these political changes “made necessary negotiations possible in the legislature” and “helped achieve consensus politically” [21], while another noted that each of these leaders had “very close access to President López Obrador” [17].

### 7.2. Pressure Group Campaigns

While the new government came in with a fresh approach and openness to address various public health and societal issues [17], health advocates still had to meet with policymakers and government officials to convince them of the importance of addressing tobacco use [19]. Interviewees again discussed the value of informing new policymakers about tobacco control and identifying key allies where they could “plant seeds” for future assistance once tobacco control initiatives reached the political agenda [17,25]. One government official claimed health advocates “developed close relationships right away with the new government” [25]. In particular, health advocates identified and worked closely with Undersecretariat López-Gatell and Deputy Medel, who were immediately interested in the topic and eager to work with health groups and convince policymakers to put tobacco control on the political agenda [26]. All interviewees claimed that the decades’ worth of evidence collected, successful regional experiences, and an increased focus on prevention were packaged in policy briefs and reports that were used to provide technical support to government officials [17,18,19,20,21,22,23,24,25,26,27,28,29,30]. This initially consisted of health advocates providing technical and legal support to Undersecretariat López-Gatell and Deputy Medel, who would be in position to further lobby “within” the administration and congress about the importance of tobacco control [21]. As a result, they organized several sessions with policymakers, especially health committee members, and government officials, especially from the Economic Ministry, aimed at convincing them that tobacco reforms would be good for the country [21,26]. Using material prepared by health advocates and inter-governmental organizations such as PAHO, both repeatedly warned policymakers about tobacco industry interference, referenced international success stories and progress, and focused on prevention, especially among youth [21,22,26]. They also cited strong public support for several tobacco control measures including 90% public support for 100% smoke-free environments in Mexico, which were well received by policymakers, especially those in the health committee [21,22,26].

Meanwhile, health advocates simultaneously operated outside these channels to directly inform policymakers with similar well-coordinated framing and messaging. This initially focused on briefing policymakers about WHO FCTC Article 5.3, which rejects tobacco industry partnerships and participating in tobacco control policymaking [26]. A few interviewees mentioned that on separate occasions, there were members of the Health Ministry and policymakers within both chambers of congress that had planned to meet with industry officials or were invited to industry-sponsored events but rejected these invitations after they were briefed about WHO FCTC Article 5.3 [17,26]. Health advocates also used regional reports and other evidence to stress the increasingly strong public support, a rise in youth consumption that was preventable, and that while Mexico was the first country in the region to ratify the WHO FCTC, it was falling behind other countries [19,23,27]. One advocate stated “all of this helped to establish the urgency” of addressing tobacco use [23], while one government official explained, “civil society put pressure from the outside, while we lobbied on the inside but we were coordinated with the same messaging throughout the process” [26].

### 7.3. Industry Interference

The tobacco industry was also very active in attempting to meet with and lobby various policymakers and government officials to prevent any tobacco control legislation from surfacing. Policymakers and government officials stated that they had found out later that industry officials had attempted to meet with newly elected policymakers, offering them gifts, vacations, and financial support in return for rejecting any tobacco control initiative to be introduced and discussed [19,28]. While these industry efforts ultimately failed, interviewees claimed that the industry was aggressive as always, and if not for a majority in MORENA policymakers in the health committee, tobacco control legislation may have not moved, thereby reiterating the importance of political and administrative change [19,28].

## 8. Decision Window: Introduction and Adoption of LGCT Reforms

As the three streams (problems, policies and politics) aligned, there emerged an opening of a policy window to finally couple everything together to put tobacco control initiatives on the political agenda and eventually amend the LGCT. This final section examines the policymaking process of tobacco control entering the political agenda in 2019 and ending with the amendment of the LGCT in 2022. 

### 8.1. Entering the Political Agenda 

According to interviewees in late 2019 and early 2020, tobacco control appeared to not only enter the political agenda but also receive significant support and attention as a priority from President López Obrador. Health advocates often engaged with President López Obrador’s advisory team through close relations with the Undersecretariat Dr. Lopez-Gatell [27]. Several interviewees claimed that tobacco made its way onto the political agenda because they could reach the president through his morning press conferences [17,19,23]. In particular, health advocates would work with journalists to ask direct questions to the president or president’s secretary, which “forced them to address the issue” and helped “position the topic of tobacco on the political agenda” [17,19]. This motivated other journalists to ask questions, which sparked further coordination with journalists through built trust, leading to tobacco control being discussed in various media outlets and “reaching the highest level of the political agenda” [19,27]. One advocate mentioned they tailored their messaging along with President López Obrador, stating, “if the president said there was ‘inequality’ then we focused on the inequality of smokers” [17].

### 8.2. Introduction of Initiatives

While tobacco control initiatives had been discussed in the past, the origin and initial draft of the eventually approved legislation to amend the LGCT was introduced in the Chamber of Deputies health committee on 3 March 2020 by Deputy Manuel Huerta [33] (Table 1). This initial draft included several provisions that aligned with the FCTC, including 100% smoke-free environments and a TAPS ban, among others. While this initial initiative was supported by health advocates, it did not gain traction in the health committee. Between September and November 2020, a series of tobacco control initiatives were introduced first by Deputy Medel and then by Deputy Frida Esparza, which again were supported by health advocates and attempted to align with the FCTC [34]. While all these bills were discussed in the health committee, none of them were voted on and, as a result, they remained pending in committee and not a priority. A few interviewees stated that the health committee president at the time, Deputy Miroslava Sànchez Galván, was not as supportive and, similar to the past, was more like a gatekeeper despite increased political will and support from the MORENA party and López Obrador administration. 

Despite these political setbacks, on 10 February 2021, Deputy Medel was appointed Chamber health committee president. Several interviewees agreed this was another key turning point that further strengthened their position and provided another opportunity for the political window to open [17,18,19,23,24,25,26,27,28]. As committee president, Deputy Medel wasted no time in holding public sessions, attempting to secure a vote in committee on some piece of legislation that amended the LGCT. Throughout the discussions of various initiatives, each of the provisions were carefully examined. Several interviewees claimed that given past experiences, there was a consensus to prioritize 100% smoke-free environments and a TAPS ban [23,24]. Other provisions including tobacco standardized packaging and a tobacco flavor ban were supported but reached strong resistance among policymakers [28]. Given previous political struggles, intensive negotiations, and a closing political window opportunity, policymakers with support of health advocates decided to drop these other provisions and prioritize 100% smoke-free environments and a TAPS ban [19,23]. Some interviewees claimed the other provisions had “less support” [23], were “too risky” [18], and acknowledged that negotiations can be quite difficult and sometimes you have to “prioritize” and “sacrifice” other parts to ensure that the main components can advance forward [18,22,23,25,27]. One government official stated, “sometimes you cannot win everything” [18], while another stated, “we needed this to advance otherwise we may have missed the political opportunity” [19]. Following these changes, on 25 March 2021, the health committee unanimously voted to approve Deputy Medel’s LGCT amendment bill [35].

### 8.3. Pressure Group Campaigns

Throughout the legislative process, health advocates continued to relentlessly support the LGCT amendment through further coordination and key communication messaging, developing a paid media campaign, producing earned media through advocacy campaigns and participating in key political activities.

#### 8.3.1. Continued Coordination and Key Communication Messaging

Health advocates developed a communication strategy with compelling and engaging messages that was supported by increased collaboration and coordination [17,24,25,26]. Strong coordination consisted of planning and consensus building internally and then framing these issues with consistent arguments and messaging repeatedly across different platforms [19]. In particular, when industry actors or opposing policymakers argued against or acted in opposition to the policy proposals, health advocates were quick and coordinated in their responses. This included supporting policymakers when they asked for advice and then communicating and coordinating which expert(s) were best positioned within the advocacy network to respond and then provide the correct political, economic, and legal advice necessary to address the opposition [17,29]. One policymaker claimed to receive a lot of help from both local and international health groups and stated “without their help we would have not done anything” and “they always approached me wanting to fight this issue [tobacco] together” [28]. Another policymaker stated, “the truth is, they never left me alone and that was great help” [26]. These policymakers further went on to state that they appreciated how the transnational tobacco control network was very coordinated with many years of experience, would help to intervene at every possible moment, and offered recommendations and assistance quickly and concisely [26,28].

Several interviewees also claimed that COVID-19 played a key role as a significant focusing event that contributed to the success of LGCT reforms [17,21,22,24,26,28]. This initially began with health advocates internally collecting and reviewing data and connecting COVID-19 to smoking by demonstrating that smoking was a key risk factor in contracting COVID-19. Health advocates then worked closely with Undersecretary Lopez-Gatell to produce key messaging and frame the tobacco issue around COVID-19 to President López Obrador, who further supported this in morning press conferences [17]. Other interviewees claimed this had a ripple effect as government announcements continued to highlight how COVID-19 was worse for those who smoke, which helped further position tobacco control support from more policymakers and journalists [22,24,26]. One advocate stated this “helped institutionalize the issue as they wanted to engage with us more” [19]. While some interviewees did not believe COVID-19 played a big role in helping amend the LGCT, they acknowledged that COVID-19 did allow them to have more direct access to policymakers through virtual zoom meetings [19,23]. Other advocates claimed this access allowed them to be more informed about policymaker positions, more connected across virtual meetings, and more coordinated in their messaging [19,21].

#### 8.3.2. Media Advocacy

Health advocates participated in several paid and earned media campaigns to help support the LGCT reforms. This consisted of paid advertisement spots in various media platforms including newspapers, television, radio, billboards, and social media. For example, local advocacy group Reflecciona, with CTFK support and participation of the entire transnational tobacco control network through Mexico SaludHable, implemented an advertising campaign called “Es por todos” (It is for everyone), which consisted of three phases: (1) raising awareness to amend the LGCT (January–May 2020), (2) promoting the reforms in congress (June 2020–June 2021), and (3) creating community support and triggering conversations on social networks (May 2021–December 2021) (Figure 1) [36]. The campaign produced powerful images, rich infographics, and captivating messages focused on youth that were employed in various media platforms including newspapers, radio, television, billboards, subway stations, buses, and social media (Figure 2). Furthermore, some of these advertisements exposed industry interference and targeted policymakers to help support passage of the LGCT reforms. This included tagging policymakers on Twitter and posting social media graphics with their faces asking them what side they were on (public health or with industry), and distributing personalized boxes with pictures of their faces (Figure 3). Other efforts consisted of gaining earned media coverage, which included the morning president press conferences as already discussed and holding press conferences of their own at critical junctures of the legislative process. For example, on 27 January 2021, local health group Salud Justa with support from CTFK and PAHO organized a press conference titled “South America is already 100% smoke-free, what about Mexico?”, which created further urgency and helped propel the Medel bill to be finally introduced in the health committee in March 2021 [37]. On 16 March 2021, Salud Justa organized another press conference to denounce PRI Deputy Fernando Galindo for colluding with the tobacco companies to delay the vote in the health committee [38]. Five days later, health groups also joined Deputy Medel in another press conference to further denounce the Chamber of Deputies economic committee for not collaborating in the discussion of amending the LGCT [39]. These efforts led to several published articles, interviews, and op-eds that helped the health committee unanimously vote to approve the Medel bill on 25 March 2021 [40].

#### 8.3.3. Political Activities

Health advocates continued to participate in public hearings and debates and proactively identify and educate policymakers to support the LGCT reforms. Understanding that the eventual initiatives may reach the senate, health advocates identified and worked closely with Senate Health Committee member Ernesto Pérez Astorga, who became another political champion to help support and facilitate the approval of the LGCT amendment in the senate [19,22,28]. One health advocate recalled that despite being a strong businessman supportive of business practices, Senator Pérez Astorga told him “that it was better to defend the right to health because it was for the general well-being than to defend the economic right of a few” [23]. In August 2021, a month before the senate resumed sessions, health advocates also worked proactively to send senators the Healthy Legislative Agenda, which included a petition to the senate to approve the LGCT reform [41]. Health advocates continued to use data from surveys, regional reports, and local subnational progress in Mexico to create infographics, fact sheets, and policy briefs on smoking prevalence rates, especially among youth, the benefits of smoke-free environments and banning TAPS, and conflicts of interest with industry, among others, to educate policymakers [19,22,28]. On 13 October 2021, health advocates participated in a senate virtual forum hosted by Senator Pérez Astorga, in which they continued to express their support for the LGCT amendment [42]. Health advocates followed up with this session, sending letters to senators in late October further requesting that the senate approve the LGCT amendment [43]. These efforts led to the Senate Health Committee also unanimously approving the LGCT amendment on 3 November 2021 and the senate legislative studies committee approving it on 2 December 2021.

### 8.4. Industry Interference

Throughout the political process, the industry continued to relentlessly oppose LGCT reforms. The tobacco industry continued to lobby policymakers to prevent or delay the process of amending the LGCT. Tobacco companies and restaurant associations also continued to iterate that the LGCT amendment would result in important economic losses for the country, including loses in economic revenue and impacting employment, as tobacco companies argued they were an important source of employment, providing thousands of jobs to workers, most notably tobacco farmers [17,18]. According to interviewees, the officials that appeared to be coopted by the industry seemed to mostly regurgitate industry economic talking points [19,22,28]. Tobacco companies also leveraged support from policymakers and tobacco farmers from tobacco-growing states such as Nayarit and Veracruz, who were also vocal throughout the process in opposing the LGCT reforms [24]. However, according to interviewees, this opposition did not gain much traction because the industry used the same recycled arguments from the past, so the health advocates were well prepared in their counterarguments [18,23]. This included arguing that most of the tobacco in Mexico is imported from other countries [18], thousands of cigarettes are produced every hour, which is more technical and not really using labor [18,24], and the tobacco that is grown in Mexico relies on industry taking advantage of child labor, paying them low wages and exposing them to pesticides that also contaminate the environment [18,23,24].

Given the industry’s struggle and failure to halt the progression of the LGCT, the industry resorted to scare tactics and threatened health advocates and government officials. This included standard legal threats to initially oppose standardized packaging proposals, but as the Medel bill focused on a TAPS ban, industry legal threats centered on how a TAPS ban was a violation of the Mexican constitution and international trade agreements [17]. This also included personal threats that largely came after a health advocacy opinion letter to the health committee that was shared with the economic committee was leaked to the industry and vaping associations [26]. Personal attacks were first launched against health advocates, claiming that particular members of international health groups were interfering with Mexican legislation, calling it a “form of colonialism” [17]. Further personal attacks were launched against policymakers and their assistants, as one policymaker claimed that she received threats from industry groups that had pictures of her son [19]. Another policymaker claimed that industry groups held demonstrations outside of their offices, arguing they did not represent them and were taking jobs away [28]. When these threats did not gain much ground with policymakers, they threatened their legislative assistants and their families [28]. Despite these personal threats and attacks to health advocates, policymakers, and their assistants, they all powered through and maintained their support for the Medel bill.

### 8.5. Perfect Storm

Following a decade of attempts to reform the LGCT, the Mexican government finally approved the amendment on 14 December 2021, which was published in the official gazette on 17 February 2022 [13]. When asked why the LGCT was finally amended, all interviewees claimed that it was a combination of continued and increasingly coordinated health advocacy support to define and frame the problem to policymakers and the media (problem stream), continued efforts to refine and modify proposals for support (policy stream), and strong political will (political stream) that provided a window of opportunity to introduce and amend the LGCT [17,18,19,20,21,22,23,24,25,26,27,28,29,30]. Several interviewees claimed that all of these elements and efforts came together to create a “perfect storm” [17,18], which allowed the amendment to be introduced and then ultimately passed. Furthermore, some claimed that even though political will is key, “you still need to take advantage of the political opportunity” [25], “you have to be prepared and coordinated for when the moment does arise” [25], and years of experience and increased coordination and collaboration, which “can take many years to build” [23], significantly helped to accomplish this.

## 9. Discussion

The Mexican case study illustrates how the persistent effort of the policy entrepreneurs resulted in the convergence of the problem, policy solution, and political process streams at the policy window. The policy window, opened by an institutionalized event (presidential election) and the activities of the persistent health advocates, ultimately led to policy adoption.

The case of Mexico further illustrates the importance of a transnational tobacco control network comprising local health groups, international health organizations, and inter-governmental organizations collectively working together to help adopt FCTC-based policies. Similar to other tobacco control case studies in LMICs [12,44], the financial and technical support from international groups helped fuel and support local efforts. Unlike other case studies that typically provide short snap shots of successes and failures during the policy process [10,11], this study examined these efforts over time, demonstrating the growth of this network from individual and isolated efforts to increased coordination and collaboration. While resources continue to play a critical role in supporting local health advocacy efforts, long-term knowledge and experience [45], as showcased in Mexico, can further institutionalize and sustain local efforts [46].

The Mexican case also illustrates the importance of subnational efforts in contributing to the success of national policy reforms. Similar to other larger countries with federally structured governments, it can be quite challenging to make federal changes, especially amidst strong tobacco industry opposition [47]. Despite failed efforts at the national level throughout the 2010s, significant progress was made at the state and local levels to produce effective subnational tobacco control policies that helped change social norms, a key ingredient that has been shown to alter public attitudes surrounding tobacco use and contribute to nationwide tobacco control policy efforts [48]. This case study is a reminder that local governments remain important laboratories for experimentation and advancing tobacco control that can have a profound effect on scaling-up efforts nationally [49]. 

The Mexican case study also highlights the importance of seizing political opportunities when they arise and operating through short political windows to adopt WHO FCTC-based policies. Similar to other successful facilitators of WHO FCTC policy adoption [12,44], health advocates were coordinated and proactive to identify and target potential supportive policymakers before political opportunities opened. This included developing close relations with policymakers in key positions who later became political champions to leverage support for tobacco control initiatives, a key facilitator in policy adoption [12,44]. This approach also recognized the importance of political timing and ensuring the initiative was approved in the health committee during a tight political window; otherwise, this could have been a missed political opportunity as demonstrated in other failed cases [50]. With the recent election of MORENA presidential candidate Claudia Sheinbaum, there could be an important political opportunity to accomplish this given the political party’s recent leadership in tobacco control.

### Limitations

Although we reached out to 24 individuals, only 14 agreed to be interviewed. Furthermore, most of the interviewees work in public health, so there is an inherent bias towards efforts and support to improve public health. However, a strength of this paper is the diversity of viewpoints, as interviewees included representatives from academia, advocacy, and government. These interviewees were also integrally involved in the policy process and provided an added level of insight to help contextualize our documented findings. 

## 10. Conclusions

The Mexican experience illustrates the importance of continued health advocacy and political will in adopting FCTC-based policies. Other countries should follow Mexico’s lead by collecting and sharing data through coordinating efforts in order to be prepared to seize political opportunity windows when strong political will is present.

## Figures and Tables

**Figure 1 ijerph-21-00917-f001:**
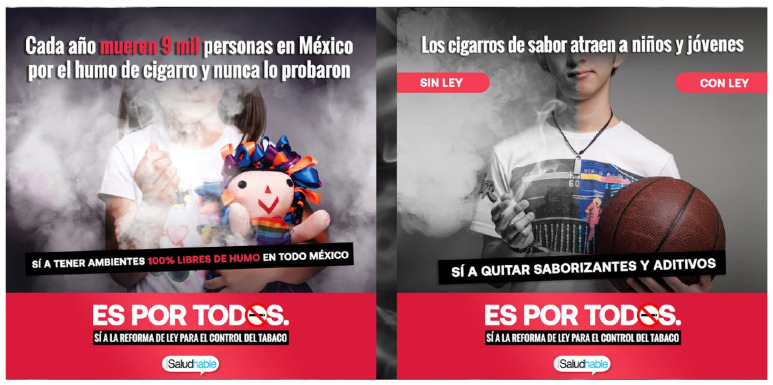
Community support and triggering conversations on social networks (May 2021–December 2021). Left side caption: Every year 9 thousand people die in Mexico from cigarette smoke and they have never tried it: Yes to having 100% smoke-free environments throughout Mexico. It’s For Everyone: Yes to the reform of the tobacco control law. Right side caption: Flavored cigarettes attract children and young people. Without the law, with the law. Yes to removing flavorings and additives. It’s For All: Yes to the reform of the tobacco control law.

**Figure 2 ijerph-21-00917-f002:**
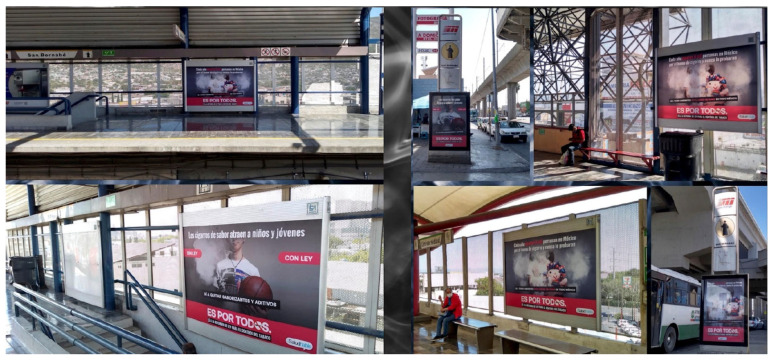
The campaign produced powerful images, rich infographics, and captivating messages focused on youth that were employed in various media platforms including newspapers, radio, television, billboards, subway stations, buses, and social media.

**Figure 3 ijerph-21-00917-f003:**
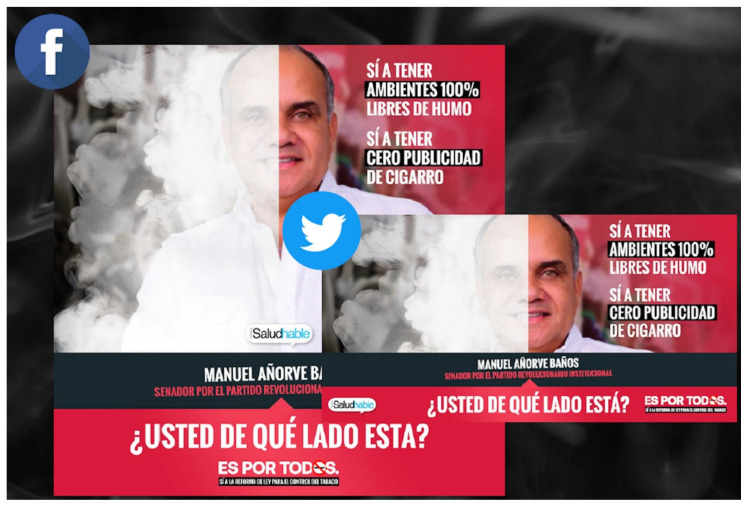
Tagging policymakers on Twitter and posting social media graphics with their faces asking them what side they were on (public health or with industry), and distributing personalized boxes with pictures of their faces. Caption: Yes to having 100% smoke-free environments. Yes to having zero cigarette advertising. Manuel Añorve Baños, Senator for the Institutional Revolutionary Party. Which side are you on? It’s For Everyone: Yes to the reform of the law for tobacco control.

**Table 1 ijerph-21-00917-t001:** Approving reforms to the General Tobacco Control Law in Mexico (2020–2022).

Date	Event
3 March 2020	Deputy Manuel Huerta presents an initiative to reform the LCGT, which includes establishing 100% smoke-free spaces and banning TAPS
19 March 2020	México Salud-Hable supports reforms to the LGCT
30 April 2020–1 September 2020	Congress recess period
19 June 2020	México Salud-Hable raises concerns about smoking and vaping and COVID-19 and urges policymakers to reform the LGCT
31 August 2020	The INSP’s Tobacco Research Department raises concerns about smoking and vaping and COVID-19 and urges policymakers to reform the LGCT
23 September 2020	Deputy Carmen Medel introduces an initiative to reform the LGCT, which includes establishing 100% smoke-free spaces and banning TAPS
17 November 2020	The Health Committee of the Chamber of Deputies holds an opinion forum, hearing from civil society organizations in support of the reforms to the LGCT
18 November 2020	Deputy Frida Esparza presents an initiative to reform the LGCT, which includes establishing 100% smoke-free spaces and banning TAPS
15 December 2020–1 February 2021	Congress recess period
27 January 2021	Salud Justa holds a virtual press conference urging policymakers to reform the LGCT
10 February 2021	Deputy Carmen Medel is appointed as president of the Health Committee of the Chamber of Deputies
18 March 2021	The Health Committee of the Chamber of Deputies holds an opinion forum, hearing from health advocacy organizations in support of the reforms to the LGCT
16 March 2021	Salud Justa holds a press conference denouncing Deputy Fernando Galindo and the PRI for collusion with tobacco companies to delay initiatives
16 March 2021	The Economic Committee of the Chamber of Deputies holds a meeting with tobacco and vaping companies and restaurants who oppose the LGCT reforms arguing it is excessive and unfair, will hurt the economy, and create unintended consequences such as creating a rise in the black market and increasing use among minors
23 March 2021	Deputy Carmen Medel holds a press conference with other Deputy members of the Health Committee, denouncing the Economic Committee for not collaborating in the discussion of the law reform and holding separate meetings with industry
25 March 2021	The Health Committee of the Chamber of Deputies unanimously votes on the Medel initiative that establishes 100% smoke-free spaces and bans TAPS
26 March 2021	Health advocacy groups issue a press release on Deputy Fernando Galindo and the PRI for violating WHO FCTC Article 5.3 for colluding with tobacco companies to delay the policymaking process
8 April 2021	Health advocacy groups send a letter to the members of the Chamber of Deputies urging them to vote on the LGCT reforms
12 April 2021	Health advocacy groups hold a press conference to pressure Deputies to vote in plenary session of the Chamber of Deputies
20 April 2021	More than 60 health advocacy organizations send a letter to Mexican President, Andrés Manuel López Obrador, urging him to support the LGCT reform
27 April 2021	Undersecretary of Health Hugo López Gatell holds a press conference discussing the importance of the LGCT reform
28 April 2021	The Chamber of Deputies votes to approve the LGCT with 415 votes in favor
2 May 2021	Tobacco companies argue to the media that the LGCT reform violates the freedoms of individuals and businesses
27 May 2021	Health advocacy groups present a report to the media on the current state of tobacco consumption in Mexico in an effort to support the LGCT reform
31 May 2021–2 September 2021	Congress recess period
24 August 2021	Health advocacy groups send new congress members (Chamber of Deputies and senate) the Healthy Legislative Agenda which includes a petition to the senate to approve the LGCT reform
7 October 2021	Senator Nadia Navarro Acevedo expressed to the media support for 100% smoke-free reforms due to tobacco-related deaths
10 October 2021	Códice members participate in the program “Learning to Age” with Patricia Kelly arguing for the need to have 100% smoke-free and vape-free spaces
13 October 2021	Undersecretary of Health López Gatell supports the need to reform the LCGT during a senate forum on smoking addiction
13 October 2021	Senator Ernesto Perez Astorga hosts a virtual senate forum where health advocacy groups urge policymakers to vote for the LGCT reform.
25 October 2021	Health advocacy groups send a letter to the senate with their support for the LGCT reform showing the positive impact in case it is approved, reinforcing that this will take Mexico to the next level of fulfillment of the WHO FCTC and MOPWER
27 October 2021	Health advocacy groups hold a press conference with Senators Ernesto Pérez Astorga and Lilia Margarita Valdez of the Senate Health Committee at the senate, urging senators to vote in favor of the LGCT reform
3 November 2021	Health advocacy groups hold a virtual press conference to discuss the increase in industry interference in Mexico as reported in the Tobacco Industry Interference Index
3 November 2021	The Senate Health Committee unanimously approves the LGCT reform without changes
22 November 2021	Health advocacy groups present a survey to senators and the media showing 82% of Mexicans support banning TAPS
25 November 2021	Health advocacy groups present to the senate the increase in industry interference in Mexico as reported in the Tobacco Industry Interference Index
14 December 2021	The senate approves the LGCT reform
15 December 2021	Civil society organizations publicly praise the senate’s approval of the LGCT reform, noting the triumph over tobacco industry interference, with Mexico becoming the 9th country in Latin America to implement TAPS ban
17 February 2022	The LGCT reform is published in the Official Gazette of Mexico

**Table 2 ijerph-21-00917-t002:** Application of the Multiple Streams Framework to approving the General Law of Tobacco Control.

Category	Sub Category	Examples
Previous efforts failing to reform LGCT	Lack of political will	Majority of PAN and PRI congressmen (2008–2018) favoring economic interests over health -No public health political champions
	Tobacco industry political influence	-Holding private meetings with policymakers -Lobbying to prevent bills from leaving health committees
	Lack of health advocacy coordination	-Isolated efforts leading to lack of coordination -Lack of physical infrastructure and resources
Problem stream	Indicators/statistics	-INSP and GATS surveys detailing magnitude of smoking and rise in youth smoking -Surveys helped define problem of tobacco use
	Feedback from other policies	-Policy success in Latin America shared with Mexican government officials in Mexico and at regional health conferences -Increased urgency for Mexico who was falling behind the rest of the region
	Framing the issue	-Shift in framing from treatment to prevention with a focus on NCDs -Shift in focus on protecting youth -Framing tobacco not only as a health issue but as an economic, development, and legal issue
Policy stream	Policy entrepreneurs	-Expansion of transnational tobacco control network comprising local and international health groups and inter-governmental organizations -Extended communication and collaboration with journalists and media outlets -Successful local efforts to pass subnational 100% smoke-free laws that further enhanced capacity
	Approaching policymakers	-Strategically identifying and cultivating relationships with potential future policymakers -Identifying strong supporters and opponents
	Policy alternatives	-General support for 100% smoke-free environments and banning TAPS -Strong opposition to tobacco flavor bans and tobacco standardized plain packaging
Politics stream	Administrative or legislative turnover	-Election of President Andrés Manuel López Obrador and majority of MORENA elected officials helped place tobacco control on the political agenda -Key public health political champions identified with congressmen Dr. Carmen Medel Palma and Ernesto Pérez Astorga
	Pressure group campaigns	-Meeting and informing new policymakers about the importance of tobacco control and high public support for 100% smoke-free environments -Lobbying policymakers from the outside and working closely with Undersecretariat of Health Dr. Hugo López Gatell to lobby within government
	Industry interference	-Attempted efforts to meet with policymakers -Political donations and gifts given to policymakers
Decision window	Entering the political agenda	-President López Obrador discussing tobacco control during morning press conferences to help position tobacco control on the political agenda -Motivated journalists to ask more questions and helped build trust with health advocates
	Introduction of initiatives	-Initial initiatives failed but priority of 100% smoke-free environments and TAPS ban maintained in bill that was eventually voted on and approved
	Pressure group campaigns	-Continued coordination and key communication to provide the correct political, economic and legal advice necessary to address any opposition -Media advocacy efforts including paid and earned media including “It is for everyone” campaign -Political activities including lobbying and holding key press conferences to support LGCT reforms
	Industry interference	-Continued lobbying of policymakers -Reiterating industry economic talking points that proposal would hurt businesses and farmers -Threatening health advocates and government officials with personal and legal threats
	Perfect storm	-Increasingly coordinated health advocacy support to define and frame the problem to policymakers and the media (problem stream) -Continued efforts to refine and modify proposals for support (policy stream) -Strong political will (political stream) that provided a window of opportunity to introduce and amend the LGCT

GATS: Global Adult Tobacco Survey; INSP: the National Institute of Public Health; LGCT: the General Law on Tobacco Control; MORENA: National Regeneration Movement; NCDs: non-communicable diseases; PAN: National Action Party; PRI: Institutional Revolutionary Party; TAPS: tobacco advertising, promotion and sponsorship.

## Data Availability

Data are unavailable due to the privacy and protection of the interviewees.

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
