# Peer review of "The Perfect Storm: Applying the Multiple Streams Framework to Understand the Adoption of a WHO Framework Convention on Tobacco Control-Based Policy in Mexico"

_ijerph, 2024, doi:10.3390/ijerph21070917_

Round 1

Reviewer 1 Report

Comments and Suggestions for Authors

Dear Editor,

Thank you for the opportunity to review this manuscript. The manuscript is well written, and it provides a lot of detail on how Mexico adopted more measures of the WHO-FCTC. The authors did a thorough job and I have just a few comments-suggestions.

Material and Methods:

lines 74 and 75 “The interviewees included public health advocates (5), academic researchers (2), Mexican policymakers and government officials (5), and inter-governmental organization officials (2)”.

It is not clear to me why the authors are deciding to cite those 2 references, were those provided to grant a description of the terms?

Data analysis.

The authors did a great job describing the Multiple Streams Framework (MSF) and I am assuming that their questions revolved around these topics. However, it would be useful for them to provide example questions for each stream to give the readers a sense of what was asked during the interviews.

I suggest the authors add a figure to explain their framework to help the reader understand better the 3 streams and there the authors can include an example of the questions used for each stream.

Results

Overall: I suggest the authors include a summary table of the findings with the columns representing each stream. This would help the reader guide itself through the numerous findings. Otherwise is difficult to remember in which stream is certain piece of information located and you have a lot of relevant information.

Without compromising any identity, is it possible for the authors to consistently include if the person that is giving the statement is an advocacy, government, or academic? I feel this could add to the understanding of the different views.

Lines 153 and 154, the authors are saying: Some mentioned that previous efforts were ‘isolated’ and did not go anywhere (17). Others claimed they did not have the physical infrastructure and human resources to carry out particular operations (24).  However, there is only one citation for each statement. Please correct.

Framing the issue

Lines 202-203 the authors are talking about how the focus tended to concentrate on treatment and assisting smokers to quit. However, it is my understanding that treatments are scarce at best, and most of the time, non-existent. Please add something about that because this sentence gives the wrong impression that Mexico has a lot of treatments available.

Figures, it might be worth to have an English translation of the images at the bottom of each figure for non-Spanish speakers.

Discussion

If the authors have the data, I suggest that they mention -at least briefly- the e-cigarette ban, since it was also part of the many things that changed on the LGCT. The authors only mentioned it in the introduction, and I feel that such a relevant and controversial topic requires more discussion.

If the authors have the information on their interviews, I urge them to include something about next steps for tobacco control in Mexico.

Author Response

Dear Reviewer,

Thank you for agreeing to review the manuscript and for the helpful review. Our responses to your comments are listed below and attached thank you. 

Reviewer 1

COMMENT 1.1: Thank you for the opportunity to review this manuscript. The manuscript is well written, and it provides a lot of detail on how Mexico adopted more measures of the WHO-FCTC. The authors did a thorough job and I have just a few comments-suggestions.

RESPONSE: Thank you.

COMMENT 1.2: Material and Methods: lines 74 and 75 “The interviewees included public health advocates (5), academic researchers (2), Mexican policymakers and government officials (5), and inter-governmental organization officials (2)”. It is not clear to me why the authors are deciding to cite those 2 references, were those provided to grant a description of the terms?

RESPONSE: We apologize for the confusion. Those numbers correspond to the number of interviewees and NOT references. We have fixed this by rewriting this with n values (e.g. n=5) to avoid any confusion.  

COMMENT 1.3: Data analysis. The authors did a great job describing the Multiple Streams Framework (MSF) and I am assuming that their questions revolved around these topics. However, it would be useful for them to provide example questions for each stream to give the readers a sense of what was asked during the interviews.

RESPONSE: Yes, we have included the interview questions as an appendix supplementary file as requested by the reviewer.

COMMENT 1.4: I suggest the authors add a figure or table to explain their framework to help the reader understand better the 3 streams.

RESPONSE: We have included a table that visually captures the 3 streams as requested by the reviewer.

COMMENT 1.5: Results: Overall: I suggest the authors include a summary table of the findings with the columns representing each stream. This would help the reader guide itself through the numerous findings. Otherwise is difficult to remember in which stream is certain piece of information located and you have a lot of relevant information.

RESPONSE: See previous response to comment 1.4.

COMMENT 1.6: Without compromising any identity, is it possible for the authors to consistently include if the person that is giving the statement is an advocacy, government, or academic? I feel this could add to the understanding of the different views.

RESPONSE: This is another good suggestion. We have updated some of these statements but there are still a couple that could compromise identify so we left those ones more general.

COMMENT 1.7: Lines 153 and 154, the authors are saying: Some mentioned that previous efforts were ‘isolated’ and did not go anywhere (17). Others claimed they did not have the physical infrastructure and human resources to carry out particular operations (24).  However, there is only one citation for each statement. Please correct.

RESPONSE: Thank you for catching this and now we have added the correct references for each quote.

COMMENT 1.8: Framing the issue: Lines 202-203 the authors are talking about how the focus tended to concentrate on treatment and assisting smokers to quit. However, it is my understanding that treatments are scarce at best, and most of the time, non-existent. Please add something about that because this sentence gives the wrong impression that Mexico has a lot of treatments available.

RESPONSE: We have softened the language in this statement to recognize even though the focus was on treatment it was still underfunded and limited at the time.

COMMENT 1.9: Figures, it might be worth to have an English translation of the images at the bottom of each figure for non-Spanish speakers.

RESPONSE: We have added an English translation in the figure captions as requested. They are as follows:

Figure 1 caption: Left side caption:  Every year 9 thousand people die in Mexico from cigarette smoke and they have never tried it: Yes to having 100% smoke-free environments throughout Mexico. It's For Everyone: Yes to the reform of the tobacco control law. (Spanish: Cada año mueren 9 mil personas en México por el humo de cigarro y nunca lo probaron: Sí a tener ambientes 100% libres de humo en todo México. Es Por Todos: Sí a la reforma de ley para el control del tabaco.)

Right side caption: Flavored cigarettes attract children and young people. Without the law, with the law. Yes to removing flavorings and additives. It's For All: Yes to the reform of the tobacco control law. (Los cigarros de sabor atraen a niños y jóvenes. Sin lay, con lay. Sí a quitar saborizantes y aditivos. Es Por Todos: Sí a la reforma de ley para el control del tabaco.)

Figure 3 caption: Yes to having 100% smoke-free environments. Yes to having zero cigarette advertising. Manuel Añorve Baños, Senator for the Institutional Revolutionary Party. Which side are you on? It's For Everyone: Yes to the reform of the law for tobacco control. (Sí a tener ambientes 100% libres de humo. Sí a tener cero publicidad de cigarro. Manuel Añorve Baños, Senador por el Partido Revolutionario Institutional. Usted de qué lado está? Es Por Todos: Sí a la reforma de ley para el control del tabaco.)

COMMENT 1.10: Discussion: If the authors have the data, I suggest that they mention -at least briefly- the e-cigarette ban, since it was also part of the many things that changed on the LGCT. The authors only mentioned it in the introduction, and I feel that such a relevant and controversial topic requires more discussion.

RESPONSE: The e-cigarette ban was NOT included in the recent law. The ban has been established previously (through different legal measures). The current and new law has included the ban on use and on advertising. It’s commercialization was already banned. As a result, we have decided not to include this information.

COMMENT 1.11: If the authors have the information on their interviews, I urge them to include something about next steps for tobacco control in Mexico.

RESPONSE: We have added a couple more sentences in the discussion section discussing the importance of taking advantage of the recent election of MORENA presidential candidate Claudia Sheinbaum and the importance of properly implementing the law to avoid unnecessary delays.

Reviewer 2 Report

Comments and Suggestions for Authors

Dear Authors,

It was interesting to read about the changes and the steps that have been taken all these years in tobacco control in Mexico.

This is a well-written paper with very minor comments from me. Although the paper has a lot of extensive detail which might not be of interest to readers beyond South America or Mexico to be specific, it is an appealing and novel approach for other settings to replicate as a way of documenting the tobacco control journey.

Materials and methods

Line 75-77

Could this be clarified — if the interviewees had access to the questions before the interviews were held - Could this not have allowed some sort of bias in their responses?

Results

It would be good for the authors to include a table on the different themes and the findings as a summary of the findings. The text portion is heavy on the reader, adding a table with key details related to the themes would lighten the reading.

Line 120-121: Could the authors be more clear with what they mean when they write “Tobacco industry interference with policymakers”

Author Response

Dear Reviewer,

Thank you for agreeing to review the manuscript and for the helpful review. Our responses to your comments are listed below and attached thank you. 

Reviewer 2

COMMENT 2.1: It was interesting to read about the changes and the steps that have been taken all these years in tobacco control in Mexico. This is a well-written paper with very minor comments from me. Although the paper has a lot of extensive detail which might not be of interest to readers beyond South America or Mexico to be specific, it is an appealing and novel approach for other settings to replicate as a way of documenting the tobacco control journey.

RESPONSE: Thank you.

COMMENT 2.2: Materials and methods: Line 75-77: Could this be clarified — if the interviewees had access to the questions before the interviews were held - Could this not have allowed some sort of bias in their responses?

RESPONSE: It is typical to send interview questions ahead of time, not in terms of bias, but in terms of preparation to receive more detailed responses. Other interview studies have followed the same approach so we approached this the same way. 

COMMENT 2.3: Results: It would be good for the authors to include a table on the different themes and the findings as a summary of the findings. The text portion is heavy on the reader, adding a table with key details related to the themes would lighten the reading.

RESPONSE: This is a great suggestion. We have created a table that details each of the streams and examples for visual appeal as recommended.

COMMENT 2.4: Line 120-121: Could the authors be more clear with what they mean when they write “Tobacco industry interference with policymakers”

RESPONSE: We characterized tobacco industry interference in the sense that they were meeting with policymakers and influencing them to not allow legislative bills from being discussed and voted on. Interference is a general way to capture this but we have amended this to say “political influence” as this better captures their approach.

Reviewer 3 Report

Comments and Suggestions for Authors

The manuscript describes the opinions and analyses prepared from interview content of 14 (out of 24 invited) stakeholders in Mexico's adoption of WHO's FCTC-based national tobacco control law. The analysis was conducted using Kingdon's multiple streams framework (MSF) approach. The background and history of tobacco control in Mexico are well presented in this document. Supporting images and a listing of key historical events is also included.

My only reservation with the contents of this manuscript is that the results section (lines 122-157) feel one-sided and entirely opinionated. The authors attributed various opinions to multiple anonymous stakeholders, but it was not clear which, if any, of the opinions were contributed by the authors. Opinions of the authors should be clearly represented.

This is a high-quality, well-prepared manuscript and I have no further concerns or recommendations for improvement.

Author Response

Dear Reviewer,

Thank you for agreeing to review the manuscript and for the helpful review. Our responses to your comments are listed below and attached thank you. 

Reviewer 3:

COMMENT 3.1: The manuscript describes the opinions and analyses prepared from interview content of 14 (out of 24 invited) stakeholders in Mexico's adoption of WHO's FCTC-based national tobacco control law. The analysis was conducted using Kingdon's multiple streams framework (MSF) approach. The background and history of tobacco control in Mexico are well presented in this document. Supporting images and a listing of key historical events is also included.

RESPONSE: Thank you.

COMMENT 3.2: My only reservation with the contents of this manuscript is that the results section (lines 122-157) feel one-sided and entirely opinionated. The authors attributed various opinions to multiple anonymous stakeholders, but it was not clear which, if any, of the opinions were contributed by the authors. Opinions of the authors should be clearly represented.

RESPONSE: We understand the frustration here but these are the opinions of the stakeholders involved that were available for interviews. However, we have added a sentence in the limitations section noting that most of the interviewees work in public health so there is an inherent bias towards efforts and support to improve public health. Furthermore, we have updated some of the interview quotes to further identify who is saying what. However, some of this is a limitation as we have to protect the identify of some of those who were interviewed.

COMMENT 3.3: This is a high-quality, well-prepared manuscript and I have no further concerns or recommendations for improvement.

RESPONSE: Thank you.

Reviewer 4 Report

Comments and Suggestions for Authors

The authors report on how Mexico adopted the WHO Framework Convention on Tobacco Control (FCTC). The authors hope that their findings will help implement policies in other countries working to reduce the burden of tobacco-related disease. As reported, the manuscript is very confusing and does not highlight the real aim of the article: to understand how Mexico came to adopt a policy based on the WHO FCTC.

Some suggestions:

· Data collection: To understand the challenges of LGCT reform, the authors interviewed key stakeholders involved in tobacco control in Mexico between December 2023 and March 2024. These interviewees reported on their experiences and lessons learned from 2008 to 2020, as reported in the results? Specifying the timeframe could clarify the mismatch between the interview period and the study period.

· Data analysis: The text does not go into the specifics of how they analyzed the data within each stream.

· Problem stream: I propose to create a single Problem Stream section and subsections entitled as shown on page nine, line 362... Problems, Policies and Politics. Where possible, to facilitate reading, I suggest reducing the text and merging the various sub-sections, e.g. the paragraphs on 'Indicators/Statistics' and 'Feedback from other policies' (lines 162-195) both deal with the data collection process and its impact. They can be combined into one paragraph.

· Decision window: As above, the text could be better organized by using headings, subheadings, and bullet points to improve readability and make it easier for readers to scan and find specific information.

· Discussion: Consider adding a sentence at the beginning of the discussion section that summarizes the key findings from the case study.

· The conclusion should include information on how other countries can learn from Mexico's experience in tobacco control and what future challenges and opportunities exist for the implementation and enforcement of the LGCT amendments.

· The citation of table 1 of the text is missing

Comments on the Quality of English Language

none

Author Response

Dear Reviewer,

Thank you for agreeing to review the manuscript and for the helpful review. Our responses to your comments are listed below and attached thank you. 

Reviewer 4:

COMMENT 4.1: The authors report on how Mexico adopted the WHO Framework Convention on Tobacco Control (FCTC). The authors hope that their findings will help implement policies in other countries working to reduce the burden of tobacco-related disease. As reported, the manuscript is very confusing and does not highlight the real aim of the article: to understand how Mexico came to adopt a policy based on the WHO FCTC.

RESPONSE: We are sorry that the reviewer was not able to connect the aim to the results of the study and found it confusing. We have added a couple of sentences in the discussion to summarize our findings stating, that “The Mexican case study illustrates how the persistent effort of the health advocates resulted in the convergence of the problem, policy solution and political process streams at the policy window. The policy window, opened by an institutionalized event (presidential election) and the activities of the persistent health advocates ultimately led to policy adoption.”

We have also added in an appendix of interview questions and a detailed table to help further understand the application of the framework and the evidence that was collected, which helps summarize the findings in a more visually appealing manner for the readers.

COMMENT 4.2: Some suggestions: Data collection: To understand the challenges of LGCT reform, the authors interviewed key stakeholders involved in tobacco control in Mexico between December 2023 and March 2024. These interviewees reported on their experiences and lessons learned from 2008 to 2020, as reported in the results? Specifying the timeframe could clarify the mismatch between the interview period and the study period.

RESPONSE: We have clarified this by stating “Between December 2023 and March 2024 [interview time period], we invited via email 24 key stakeholders that had been involved in tobacco control in Mexico between 2008 and 2020 [study time period], to participate in an in-depth interview.”

COMMENT 4.3: Problem stream: I propose to create a single Problem Stream section and subsections entitled as shown on page nine, line 362... Problems, Policies and Politics. Where possible, to facilitate reading, I suggest reducing the text and merging the various sub-sections, e.g. the paragraphs on 'Indicators/Statistics' and 'Feedback from other policies' (lines 162-195) both deal with the data collection process and its impact. They can be combined into one paragraph.

RESPONSE: We have created a table that encompasses all 3 streams to address this comment and offer more visually appealing aspects of the data. We have condensed some of the writing but part of this is the framework we are using (more visual in the table). We also feel the quotes and data written provide a rich depiction of how the policy was passed. We hope the table helps address this comment.  

COMMENT 4.4: Decision window: As above, the text could be better organized by using headings, subheadings, and bullet points to improve readability and make it easier for readers to scan and find specific information.

RESPONSE: We have already labeled the headings in bold and the subheadings in bold/italic to differentiate the sections. However, we feel that the added table will help better visualize this.

COMMENT 4.5: Discussion: Consider adding a sentence at the beginning of the discussion section that summarizes the key findings from the case study.

RESPONSE: We have added two sentences at the beginning of the discussion section summarizing the main findings as recommended by the reviewer. See previous response to comment 4.1.

COMMENT 4.6: The conclusion should include information on how other countries can learn from Mexico's experience in tobacco control and what future challenges and opportunities exist for the implementation and enforcement of the LGCT amendments.

RESPONSE: In the conclusion we state “The Mexican experience illustrates the importance of continued health advocacy and political will in adopting FCTC-based policies. Other countries should follow Mexico’s lead by collecting and sharing data through coordinating efforts in order to be prepared to seize political opportunity windows when strong political will is present.” Since this paper deals with adoption of the policy our conclusions reflect recommendations for approving a similar policy as opposed to the implementation of the policy which is outside the scope of the present paper.

COMMENT 4.7: The citation of table 1 of the text is missing

RESPONSE: Thank you for noticing this. We have updated the text to include a citation for table 1, which is now table 2.

Round 2

Reviewer 4 Report

Comments and Suggestions for Authors

The authors have answered the reviewer's questions. The manuscript has been improved and is now suitable for publication in IJERPH.